# INTERPRETABLE NEURAL DECODING THROUGH DYNAMICAL EMBEDDINGS

## ABSTRACT

Decoding and forecasting human behavior from neuroimaging data is a fundamental challenge spanning neuroscience, artificial intelligence, and machine learning. Naturalistic tasks such as real-world navigation generate complex, nonlinear dynamics that are difficult to model: linear methods cannot capture these interactions, while deep learning architectures often overfit in the limited and noisy data regimes typical of fMRI. We introduce Manifold Dimensional Expansion (MDE), a simple yet powerful prediction algorithm grounded in dynamical systems theory. Leveraging the generalized Takens theorem and Simplex projection, MDE reconstructs latent state spaces directly from voxelwise fMRI signals and integrates feature selection with cross-validation to identify causally relevant neural drivers of behavior. Applied to a naturalistic driving task, MDE predicts Steering, Acceleration, and Braking from fMRI time series with accuracy comparable to or exceeding regression and Transformer baselines. Crucially, MDE is the first method to combine strong predictive performance with guaranteed mechanistic interpretability, as it does not rely on latent variables. This property enables causal insights into brain–behavior dynamics. Such interpretability is essential in neuroscience, where the goal is not only to predict but also to discover and understand the mechanisms linking neural activity to behavior, insights that are critical for advancing scientific understanding and guiding interventions. More broadly, our results demonstrate that manifold-based dynamical embeddings offer a principled path toward accurate, causally grounded forecasting of complex nonlinear systems in domains where interpretability is as important as performance.

## 1 INTRODUCTION

A fundamental test of our understanding of complex natural systems is the ability to accurately predict their future behavior from past observations. Modeling and forecasting human behavior remains a central challenge across neuroscience, machine learning, and artificial intelligence. In particular, naturalistic tasks such as real-world navigation inherently generate complex, nonlinear dynamics in neural activity. These dynamics pose significant challenges for existing modeling approaches: traditional linear methods are fundamentally limited in their ability to capture nonlinear relationships, while more flexible deep learning architectures typically require large-scale datasets and are prone to overfitting in the limited, noisy environments characteristic of neural recordings. As a result, accurately predicting behavior from fMRI data during naturalistic tasks remains an unsolved and pressing problem. Explicit mechanistic modeling is often infeasible for naturalistic tasks that unfold through continuous perception, decision-making, and action. Nevertheless, accumulating evidence indicates that task-relevant cognitive and behavioral dynamics are embedded within low-dimensional neural manifolds, even when observed in the high-dimensional space of brain activity Churchland et al. (2012); Eckmann and Tlusty (2021); Zhang et al. (2021); Fontenele et al. (2024). This observation makes state-space reconstruction a particularly well-suited approach, as it leverages the manifold structure to recover the underlying dynamics while providing both predictive power and mechanistic interpretability. In this work, we introduce **Manifold Dimensional Expansion (MDE)**, a theoretically principled algorithm for decoding and forecasting behavior from neural time series. Grounded in the generalized Takens' theorem Deyle and Sugihara (2011), MDE leverages the Simplex projection algorithm Sugihara and May (1990) prediction skill maximization via feature selection and convergent cross-mapping (CCM) causal inference Sugihara et al. (2012)

to reconstruct low-dimensional latent dynamics directly from voxel time series. This enables accurate short-term behavioral forecasting without resorting to complex or overparameterized black-box models. By explicitly identifying causally relevant features, MDE provides interpretable embeddings of the dynamical structure underlying brain-behavior relationships. We validated MDE on a naturalistic driving experiment (Fig. 1), demonstrating its ability to predict behavior one time step into the future (each step corresponding to 2 seconds) from fMRI recordings. MDE achieved accuracy comparable to or exceeding that of regularized regression and Transformer baselines, while offering a mechanistic explanation of the observed dynamics through causal feature attribution and manifold reconstruction. Although we focus on fMRI decoding, the framework is broadly applicable to forecasting any high-dimensional time series dataset with low-dimensional latent dynamics, such as financial markets, climate systems, or robotic control.

Our contributions are threefold:

- **Algorithmic:** We propose MDE, a new forecasting algorithm that integrates state-space reconstruction with causal feature selection, enabling interpretable manifold embeddings from high-dimensional time series.

- **Theoretical:** MDE is grounded in dynamical systems theory, providing principled conditions (via Takens' theorem and CCM) for reconstructing low-dimensional dynamics and attributing causal drivers.

- **Empirical:** On naturalistic fMRI data, MDE matches or outperforms strong baselines (Lasso, Ridge, Transformers) while uniquely offering mechanistic explanations of brain–behavior dynamics, highlighting its value as both a decoder and a general-purpose forecasting tool.

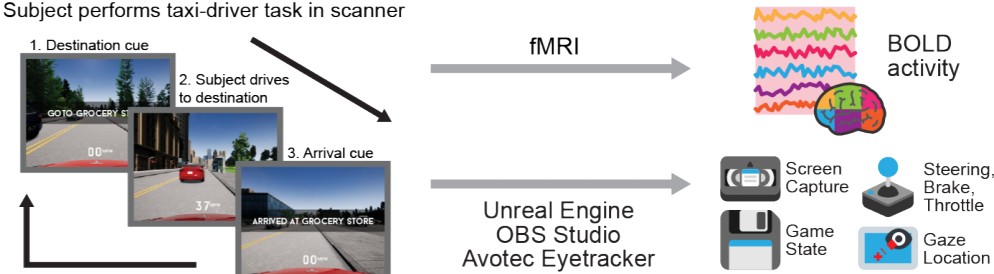

Figure 1: Experimental setup: naturalistic behavior in fMRI. The subject performed a taxi driver task while brain activity was recorded with fMRI. On each trial, the subject was instructed to navigate to a randomly-selected destination while obeying all traffic rules. Ground-truth behavioral data was recorded using a combination of the built-in Unreal Engine recording system, OBS studio, and an Avotec eyetracker.

## 2 RELATED WORK

Early approaches to fMRI decoding and analysis primarily relied on linear regression to link experimental features with brain activity Friston et al. (1994); Kriegeskorte et al. (2008); Naselaris et al. (2011). These methods have been instrumental in mapping functional regions and representational structures, but they are inherently limited in capturing nonlinear relationships between brain activity and behavior. Extensions based on engineered feature spaces, such as spatiotemporal Gabor filters for vision Nishimoto et al. (2011) or embeddings from language models for semantic processing Huth et al. (2016); Tang et al. (2023), partially address these limitations but still depend on linear mappings. Moreover, studies of naturalistic behaviors, such as driving Spiers and Maguire (2007); Choi et al. (2017); Mader et al. (2009), have largely provided correlational insights rather than predictive or mechanistic explanations. Recent work has applied deep learning to neural decoding, including recurrent, convolutional, and autoencoder-based architectures Sussillo et al. (2012); Pandarinath et al. (2018); Tseng et al. (2019); Livezey and Glaser (2021); Zhou and Wei (2020). These

models capture nonlinear dynamics and often achieve strong predictive performance. Transformer-based models have also shown promise in fMRI decoding tasks Nguyen et al. (2020); Zhao et al. (2022); Candelori et al. (2024). However, such models are typically data-hungry, prone to overfitting in noisy, small-sample regimes, and difficult to interpret, characteristics that limit their utility in neuroimaging and other high-dimensional scientific domains. A complementary perspective is to model neural activity as a dynamical system. This line of work reconstructs latent state spaces that capture the temporal structure of neural processes Churchland et al. (2012); Abbaspourazad et al. (2021); Schneider et al. (2023), with reviews emphasizing their potential for neuroimaging John et al. (2022). While powerful for uncovering low-dimensional dynamics, many of these methods rely on hidden-variable models that prioritize compactness over interpretability, making it difficult to connect causal neural features to behavior. Our work bridges these lines of research by introducing a forecasting framework that combines the predictive strength of machine learning with the mechanistic interpretability of dynamical systems theory. Unlike black-box deep learning models or purely correlational linear approaches, our method explicitly reconstructs neural dynamics from high-dimensional time series and attributes causal contributions, enabling accurate prediction together with explanatory insight.

## 3 METHOD: MDE AND DYNAMICAL SYSTEMS FOUNDATIONS

In this work, we address the task of predicting behavioral outputs from brain activity recorded by fMRI during a naturalistic navigation task. The primary challenge lies in inferring meaningful, low-dimensional latent dynamics from high-dimensional, noisy brain activity recordings. The foundation of our method lies in Takens' theorem, a key result in dynamical systems theory that provides the conditions under which the latent state space of a deterministic system can be reconstructed from a sequence of observations Takens (1981a). Given a time series $x(t)$, Takens demonstrated that one can construct a state-space embedding using delayed values of the observed variable:

$$\mathbf{y}(t) = [x(t), x(t - \tau), x(t - 2\tau), \ldots, x(t - (E-1)\tau)] \; (univatiate \; delay \; embedding)$$

where $E$ is the embedding dimension and $\tau$ is the time delay. If the embedding dimension satisfies $E \geq 2D + 1$, where $D$ is the true dimension of the underlying attractor, then the reconstructed space is diffeomorphic to the system's original state space, meaning it preserves the same dynamical and topological properties. This ensures that the attractor can be faithfully recovered from partial observations, which is especially valuable in fMRI studies where only a limited set of neural measurements is available. Takens' generalized theorem (Deyle and Sugihara, 2011) further shows that state-space reconstruction can be achieved from multiple observables rather than delays of a single one:

$$\mathbf{y}(t) = \big[x_1(t), x_2(t), \ldots, x_E(t)\big] \; (multivariate \; embedding)$$

where $\{x_i(t)\}$ are distinct time series (e.g., voxel signals) and $E$ is the embedding dimension (see Appendix for details). This is especially important for fMRI data, where each voxel signal captures a distinct projection of the latent neural dynamics. By integrating several signals into a multivariate embedding, we can reconstruct a more complete representation of the system's evolution over time, recovering neural trajectories without relying on latent variables or parametric models. The reconstructed state spaces provides the basis for accurate forecasting of behavior. To forecast behavior from the reconstructed state space, we adopt Simplex Projection algorithm, a nonparametric prediction method developed by Sugihara and May (1990). This technique uses the local geometry of the embedded space to produce short-term forecasts. Given an embedded point $\mathbf{y}(t^*)$ at time $t^*$, we locate its $E + 1$ nearest neighbors in the embedding space. The future state $x(t^* + \Delta t)$ is then predicted as a weighted average of the future observations associated with these neighbors. The weights are computed using an exponential decay function based on Euclidean distances, ensuring that closer neighbors contribute more significantly to the prediction:

$$\hat{x}(t^* + \Delta t) = \sum_{i=1}^{E+1} w_i \, x(t_i + \Delta t), \quad w_i = \frac{e^{-d_i/d_{\min}}}{\sum_{j=1}^{E+1} e^{-d_j/d_{\min}}},$$

where $d_i$ is the distance between $\mathbf{y}(t^*)$ and its neighbor $\mathbf{y}(t_i)$, and $d_{\min}$ is the smallest of these distances. Simplex Projection offers several advantages: it preserves local nonlinear structure, does not require an explicit parametric model, and avoids overfitting by relying on geometric regularities in the data. These features make it well-suited for analyzing complex, nonlinear dynamical systems such as neural dynamics inferred from fMRI.

## 3.1 The MDE Algorithm

MDE is a feature selection and prediction framework designed for time series forecasting based on state-space reconstruction. It is tailored to identify subsets of features (e.g. voxels time series) that are both causally relevant and predictive of behavior, leveraging principles from nonlinear dynamics and causal inference. At its core, the method uses the Simplex Projection algorithm implemented in the pyEDM library to forecast future behavioral states from reconstructed embeddings. It follows a forward feature selection strategy, incrementally building multivariate embeddings by adding voxel features one at a time. The only parameters used are the embedding dimension $E$ and time delay $\tau$ of the target variable that are automatically estimated from the training data, unless explicitly specified by the user. Details on how these parameters are estimated within the MDE algorithm can be found in the Appendix.

## 3.2 Pseudocode for MDE Algorithm

---
**Algorithm 1** MDE

---
**Require:** fMRI time series $data$, target; horizon $T_p$; maximum size $d_{\max}$; # folds $K$, convergence check flag $conv$
 1: Split $data$ into $train$ and $test$ (preserve temporal order)
 2: **for** $k = 1..K$ **do** ▷ CV on the training portion
 3:     Split $train$ into ($train_k$, $val_k$) (preserve temporal order)
 4:     Estimate target embedding $(E, \tau)$ on $train_k$ (if not provided)
 5:     $S_k \leftarrow \emptyset$ ▷ selected voxels for fold $k$
 6:     **First voxel (univariate)**
 7:     For each voxel $v$ in $data$:
 8:       Build *univariate* delay embedding of $v$ with $(E, \tau)$
 9:       Predict the target variable using **Simplex projection** and compute prediction score
10:       Run **CCM**($v \rightarrow$ target); require *convergence*
11:     Let $v^{(1)}$ be the highest-scoring voxel among those that pass CCM:
12:     $S_k \leftarrow \{v^{(1)}\}$
13:     **Subsequent voxels (multivariate)**
14:     **while** $|S_k| < d_{\max}$ **do**
15:       $\mathcal{C} \leftarrow$ all voxels not in $S_k$;
16:       For each $v \in \mathcal{C}$:
17:         Build *multivariate* state space from $S_k \cup \{v , v^{(1)}\}$
18:         Predict the target variable using **Simplex projection** and compute prediction score
19:         Run **CCM**($v \rightarrow$ target); require *convergence*
20:       Among CCM-passing candidates, let $v^\star$ be the best-scoring
21:       $S_k \leftarrow S_k \cup \{v^\star , v^{(1)}\}$
22:     **end while**
23:     **Fold evaluation**
24:     Build multivariate state space from $S_k$; train **Simplex** on $train_k$; evaluate on $val_k$ to get fold score $V_k$
25: **end for**
26: **Final test**
27: Choose a final voxel set (frequency-based)
28: Train **Simplex** on full $train$ with chosen set; evaluate on $test$; report MAE, RMSE

---

The algorithm integrates feature selection with $K$-fold cross-validation to evaluate generalization performance. In each fold, it incrementally constructs a multivariate embedding by selecting up to a user-specified maximum number of features, $d_{\max}$ (set to 50 in this work). At each selection step, candidate voxels are ranked by their predictive accuracy when added to the current embedding, as measured with Simplex projection on the training and validation split. The top-ranked voxel is then subjected to a Convergent Cross Mapping (CCM) test Sugihara et al. (2012) to verify that it exerts a causal influence on the target variable; only voxels that pass the CCM convergence criterion are retained. This procedure is repeated until either $d_{\max}$ voxels have been selected or no additional voxel both improves prediction and passes the causal test. This two-stage selection process ensures

that each voxel included in the final embedding contributes mechanistically, not just statistically, to behavior prediction. After cross-validation, the final feature set is constructed by aggregating selections across folds (e.g., by frequency of occurrence). This set is then used to build the multivariate embedding for out-of-sample evaluation on the test data, providing a principled balance between predictive accuracy and causal interpretability. Parallel computing via `joblib` accelerates the evaluation of candidate voxels, making the method scalable to high-dimensional fMRI data. For training and evaluation, we adopt a leave-one-run-out cross-validation framework. In each iteration, one fMRI run is held out for testing, while the remaining four runs are concatenated and used for training. Within the training set, we perform a 10-fold cross-validation to guide feature selection. Model performance is evaluated on the held-out run using two complementary metrics: Mean Absolute Error (MAE), which measures the average absolute difference between predictions and ground truth; and Root Mean Squared Error (RMSE), which captures the average magnitude of prediction errors. These metrics together provide a comprehensive assessment of both predictive accuracy and consistency.

## 4 EXPERIMENTAL SETUP

### 4.1 DATA DESCRIPTION

We used Unreal Engine 4 and the CARLA plugin to build a driving simulator that contains a large $2 \times 3$ km virtual city populated by dynamic AI pedestrians and vehicles Zhang (2021). The subject learned the layout of the world prior to scanning, examples of the virtual world are provided in the Appendix Fig.4. In the MRI, the subject performed a taxi-driver task. On each trial, a destination was randomly selected from 77 possible locations across the map. A text cue was displayed at the center of the screen for 2 seconds. The subject then drove to the destination via the fastest path while obeying all traffic laws and responding to vehicular and pedestrian traffic. At the destination, the subject came to a complete stop to indicate arrival. A text cue was then displayed for 2 seconds to acknowledge their arrival. After a randomized jitter of 4-12 seconds, a new trial then began (Fig.1). More details about fMRI data acquisition and preprocessing are provided in the Appendix. Four subjects with normal vision participated in this study, performing 5 experimental runs. The experimental procedures were approved by the Institutional Review Board at the University (anonymized for review), and written informed consent was obtained from the subject. From the experiment recordings, we extracted three behavioral output dimensions at 15 Hz and then downsampled to match the fMRI sampling rate. Steer Angle: the Steer angle is recorded on a range of [-1, 1] in which 0 is the neutral position, -1 is maximum Steer to the left, and +1 is maximum Steer to the right. Acceleration: the throttle input is recorded on a range of [0, 1] in which 0 is no input, and 1 is maximum throttle. Braking: the Brake input is recorded on a range of [0, 1] in which 0 is no input, and 1 is maximum braking.

### 4.2 BASELINES

As predictive baselines, we used linear models with $\ell_1$ (Lasso) and $\ell_2$ (Ridge) regularization from `scikit-learn` (v. 1.6.1). All voxel-wise time series were $z$-scored prior to fitting. Regularization strengths were optimized via Bayesian optimization (50 evaluations) with `scikit-optimize` (v. 0.10.2), nested within 5-fold cross-validation, and all random seeds were fixed. Hyperparameters minimizing RMSE on the validation folds were selected by searching log-uniformly over $\alpha \in [10^1, 10^8]$ for Ridge and $C^{-1} \in [10^{-5}, 10^1]$ for Lasso. To assess whether restricting the regression models to causally relevant features improved prediction or interpretability, we also repeated Lasso and Ridge regression after masking the voxel space to include only those identified as causal to the behavior by CCM. This analysis allowed us to test whether access to causal features alone was sufficient for improved prediction, or whether nonlinear, manifold-based modeling (as in MDE) was required. For our deep learning baselines, we implemented seven `PyTorch` models using PCA-reduced inputs. To enable a fair dimensionality comparison with MDE, we used 50 principal components, matching the maximum number of features used by MDE. Additionally, we evaluated the same models with a number of components which capture over 99% of the variance in the data. The tested architectures included a multilayer perceptron (MLP), a bottleneck regressor, recurrent networks (GRU, bidirectional GRU, and LSTM), a temporal convolutional network (TCN), and a Transformer encoder. We did not apply causal masking to the deep learning baselines,

as these models are capable of capturing nonlinear dependencies directly from the data without requiring explicit causal preselection. Training followed a 10-fold cross-validation procedure on the concatenated training runs, combined with a leave-one-run-out strategy for evaluation. Further details on the model architectures are provided in the Appendix.

# 5 RESULTS

We evaluated MDE in a leave-one-run-out cross-validation framework. We assessed the stability of MDE performance as a function of the number of selected features ($d_{\max}$), testing it with increasing feature counts from 10 to 50 (see appendix Fig.6). The final results reported here were obtained with $d_{\max}$=50. We compared MDE against ridge regression, lasso regression, and deep learning models trained on PCA-reduced data (retaining either 50 components just like MDE, or 99% of variance). We report transformer results in the main tables and provide results for other deep models in the Appendix (Fig.7, Fig.8). Performance was assessed using mean absolute error (MAE) and root mean squared error (RMSE), Tables1 and 2 report mean and standard deviation for each model and behavioral target.

Table 1: MAE between observations and predictions for each model and target (mean ± std).

| model
target | MDE | lasso | ridge | transf-50 | transf-99 |
|---|---|---|---|---|---|
| Acceleration | 0.23 ± 0.06 | 0.21 ± 0.04 | 0.21 ± 0.04 | 0.23 ± 0.02 | 0.21 ± 0.02 |
| Brake | 0.15 ± 0.06 | 0.16 ± 0.06 | 0.16 ± 0.06 | 0.17 ± 0.02 | 0.18 ± 0.02 |
| Steer | 0.05 ± 0.03 | 0.06 ± 0.04 | 0.06 ± 0.04 | 0.08 ± 0.01 | 0.06 ± 0.01 |

Table 2: RMSE between observations and predictions for each model and target (mean ± std).

| model
target | MDE | lasso | ridge | transf-50 | transf-99 |
|---|---|---|---|---|---|
| Acceleration | 0.27 ± 0.07 | 0.25 ± 0.04 | 0.25 ± 0.04 | 0.28 ± 0.02 | 0.26 ± 0.02 |
| Brake | 0.21 ± 0.08 | 0.22 ± 0.08 | 0.22 ± 0.08 | 0.24 ± 0.03 | 0.25 ± 0.04 |
| Steer | 0.1 ± 0.05 | 0.1 ± 0.05 | 0.1 ± 0.05 | 0.11 ± 0.02 | 0.09 ± 0.02 |

For the Steer target, MDE achieved the lowest MAE ($0.05 \pm 0.03$) and tied for the lowest RMSE ($0.10 \pm 0.05$), matching or outperforming all other models. This demonstrates strong predictive accuracy and robustness for a target characterized by complex, nonlinear dynamics. For the Acceleration and Brake targets, lasso and ridge regression achieved the best overall performance, while MDE remained competitive with nearly identical error values (Acceleration: $0.23 \pm 0.06$ MAE, $0.27 \pm 0.07$ RMSE; Brake: $0.15 \pm 0.06$ MAE, $0.21 \pm 0.08$ RMSE). This indicates that while MDE consistently achieves strong performance across all targets, the advantage of manifold embedding is most pronounced for the more complex Steer dynamics that arise from low-dimensional nonlinear interactions, rather than for relatively linear or lower-variance targets.

## 5.1 INTERPRETABILITY

As shown in Fig. 2, MDE allows us to reconstruct target-specific neural manifolds and yields predictions that closely track the observed behavioral time series. Figure 3 maps the cortical voxels selected by MDE onto the brain surface for one example subject, revealing the functional networks predictive of each behavioral target. Because feature selection is sparse, it is difficult to identify the cortical extent of functional regions from selected voxels alone. To address this, we computed brain-wide correlations with the selected voxels and projected them onto the cortical surface, thresholded at the 98th percentile. Importantly, MDE identifies voxels whose activity is not only correlated but causally and dynamically coupled to the behavior, offering mechanistic insights into the neural networks underlying complex human actions.

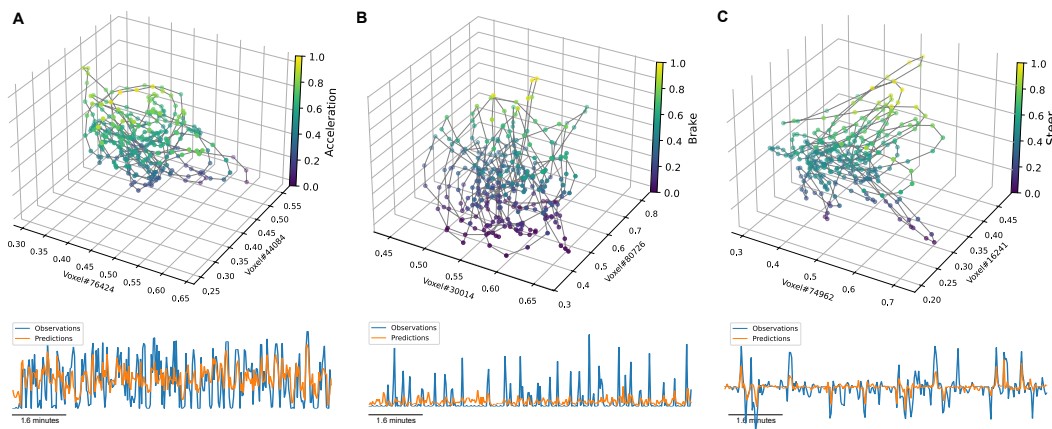

Figure 2: Neural manifolds for **A** Acceleration, **B** Brake and **C** Steer, with the vertical axis and color encoding target amplitude and the two horizontal axes showing the top two MDE-CV selected features. Below are examples target time series (blue) and model prediction (orange).

Our results reveal that each behavioral output is predicted by a distinct functional network. Acceleration is associated with dorsal premotor cortex (PM), supplementary motor cortex (SM), punctate regions in anterior lateral parietal cortex (LPC) and prefrontal cortex (PFC), the human middle temporal complex (hMT+), and the visual periphery. Brake is associated with the primary motor and somatosensory regions for the feet and hands (M1H/F, S1H/F), the frontal eye fields (FEF), the supplementary motor foot area (SMFA), and parts of the intraparietal sulcus (IPS). Steer is associated with M1H and S1H, PM, the supplementary hand motor area (SMHA) and hMT+. While the involvement of primary and supplementary motor regions is expected, these findings suggest that additional regions across association cortex are also causally involved in producing driving behavior. Taken together, these results align with known neural substrates of motor outputs while offering new insights into additional regions that may support behavior in naturalistic tasks.

As a comparison, in Appendix Fig. 9 we show that ridge regression weights for the three behavioral targets are spatially noisy and do not map onto clear networks, suggesting that random correlations may contribute to ridge performance despite regularization. Lasso produces sparser maps than ridge, but its selected voxels are scattered across the cortical surface and do not highlight behavior-specific functional networks. Both regression models achieve predictive performance comparable to MDE, but performance does not improve when restricted to voxels identified as causal (see Appendix Fig 10 and Tables 3 - 4), indicating that regularized linear models mainly exploit correlated signals sufficient for prediction, and that causality alone does not enhance performance when paired with linear decoders. Transformer models, which achieved the strongest performance among deep learning baselines, also fail to provide robust neurobiological interpretability (see Appendix Fig. 9). Voxel-level attributions are unstable across PCA settings: maps derived from PCA–50 and PCA–99% reductions differ substantially, correlations between maps remained modest ($r < 0.6$) and voxelwise sign consistency hovered between 55–70%. For Acceleration, stability was particularly low ($r = 0.26 \pm 0.07$), while Brake exhibited higher but still imperfect robustness ($r = 0.56 \pm 0.15$). This suggests that the increased sparsity observed in the PCA–99% case largely reflects variance dilution and cancellation across weak components rather than genuine feature selection.

## 6 DISCUSSION

In this work, we introduced Manifold Dimensional Expansion (MDE), a new feature selection and time series prediction algorithm grounded in dynamical systems theory, particularly generalized Takens' theorem. We applied MDE to decode and forecast human behaviors from fMRI data collected during a naturalistic driving task. The intrinsic low dimensionality of behaviorally relevant neural dynamics justifies using Takens-based reconstructions, whose assumptions align with the known structure of brain–behavior coupling. By merging theoretical insights from dynamical systems with causal feature selection, MDE effectively addresses critical limitations of existing decoding strategies.

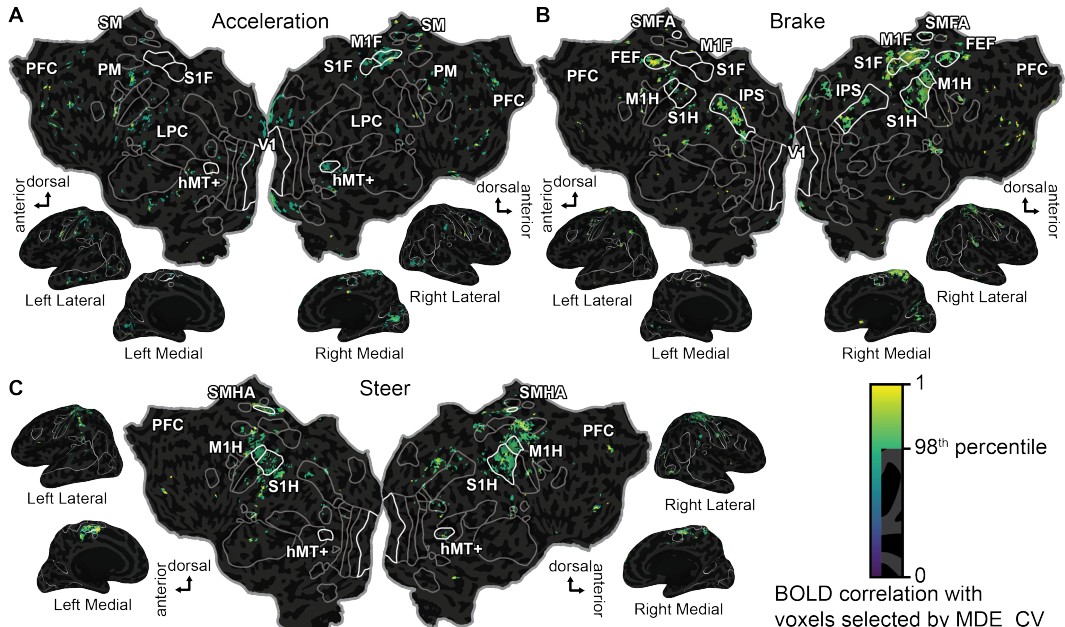

Figure 3: Features selected by MDE define task-specific networks. Brain-wide BOLD correlations (98th-percentile threshold) with these voxels are shown on flattened and inflated hemispheres for **A** Acceleration, **B** Brake, and **C** Steer. Each network engages primary/supplementary motor areas and localized prefrontal regions. ROI abbreviations in Appendix.

## 6.1 INTERPRETABILITY VERSUS PREDICTIVE SUFFICIENCY

Our comparisons with regression and deep learning models highlight a fundamental distinction between predictive sufficiency and mechanistic interpretability. Ridge regression achieved strong predictive accuracy but distributed, spatially noisy weights that show little anatomic specificity, providing little mechanistic insight. Lasso, by enforcing sparsity, selects voxels spread all over the cortical surface. It initially selected motor and somatosensory regions, consistent with their causal role in braking, but failed to consistently retain them when constrained to causal voxels. Importantly, neither Ridge nor Lasso showed improvements in predictive performance when restricted to causal features, meaning that causality alone does not enhance prediction when paired with linear decoders. Together, these findings suggest that linear regularized models exploit correlated structures sufficient for prediction without isolating the neural circuits that generate behavior, . Accordingly, the feature weights provided by these models cannot be taken as reliable indicators of mechanistic involvement. Transformer models, the best-performing among the deep learning baselines tested here, also revealed limitations for interpretability. To avoid overfitting, transformers required PCA preprocessing, which breaks the direct link between features and brain regions. When comparing voxel-level maps across PCA configurations (50 components vs. 99% variance explained), we found limited stability: average correlation was 0.4 and sign consistency of 62.6%. Moreover, the increased sparsity observed in the 99% PCA case reflected variance dilution and cancellation across weak components rather than genuine feature selection. These results indicate that while transformers leverage distributed patterns for accurate predictions, the specific regions selected are unstable and should not be overinterpreted neurobiologically.

By contrast, MDE produces sparse and stable voxel sets directly tied to causal feature selection, without reliance on dimensionality reduction. Its sparsity is qualitatively different from that observed in PCA-based models: it arises from the convergent cross-mapping (CCM) criterion, which explicitly isolates causally relevant features rather than artifacts of projection. The key contribution of MDE is to marry causal feature selection (via CCM) with a nonlinear forecasting engine (via simplex projection), enabling us to capture state-dependent and potentially chaotic dynamics that global linear maps cannot represent. This integration not only supports robust prediction of brain–behavior dynamics but also yields transparent, interpretable feature sets that preserve mechanistic insight. As

a result, MDE consistently recovers motor and somatosensory cortices as causal drivers of behavior while revealing novel prefrontal contributions, providing a mechanistically interpretable account of brain–behavior relationships. With very few features, MDE achieves predictive performance comparable to regression and deep learning baselines, and outperforms them on predicting Steer behavior (a complex, nonlinear control signal), underscoring its ability to decode behaviors where nonlinear modeling is particularly beneficial.

In addition to the primary, supplementary, and pre-motor regions, MDE identified several regions in the prefrontal and parietal cortices that may causally drive behavior. These regions may implement higher-order abstract cognitive processes that lead to concrete motor actions, and are promising areas for future study. Additionally, we find hMT+, which processes optic flow, to be predictive of Steer and Acceleration, an apparently counterintuitive result. However, human drivers saccade in the direction of upcoming turns before executing the turn Land and Lee (1994). The associated optic flow from these saccades are then directly predictive of turning actions, thus likely making hMT+ predictive of Steer. The relationship between hMT+ (and visual periphery) activity and Acceleration is less clear. We hypothesize that this may be part of a feedback control loop between optic flow perception and Acceleration control; however, this relationship would require further investigation.

## 6.2 IMPLICATIONS FOR NEURAL DECODING

Our results suggest that the dimensionality and control characteristics of driving actions shape the degree of linearity in the brain–behavior mapping: relatively simple, monotonic outputs such as Acceleration and Brake can be captured by linear models, while Steer, which requires continuous, closed-loop integration of sensory feedback and motor planning, benefits from nonlinear manifold reconstruction. Importantly, MDE avoids the extensive hyperparameter tuning, architectural complexity, and overfitting risks of deep learning approaches, providing a data-efficient method well suited to fMRI studies with limited sample sizes.

## 6.3 LIMITATIONS AND FUTURE WORK

Our approach relies on the assumption that the target time series arises from a predominantly deterministic dynamical system residing on a low-dimensional attractor. Strong stochasticity or high intrinsic dimensionality may degrade performance. Computational cost also scales with embedding dimension, but GPU implementation of the algorithm could mitigate this limitation in the future. Beyond neuroscience, MDE's framework can be applied to any partially observed dynamical system, as discussed in the Appendix section F.

## 6.4 CONCLUSION

We conclude that MDE offers a principled bridge between prediction and mechanism in neural decoding. Unlike regression and deep learning baselines, which provide predictive sufficiency without interpretability, MDE yields sparse, stable, and causally grounded voxel sets that explain behavior mechanistically. This positions MDE as a valuable methodological advancement in NeuroAI, meeting the growing demand for robust, interpretable, and data-efficient predictive models, and paving the way toward critical applications such as brain–computer interfaces where stability and causal grounding are essential.

## 7 ETHIC STATEMENT

The experimental procedures were approved by the Institutional Review Board at the University (anonymized for review), and written informed consent was obtained from all participants. The dataset is currently under review as part of a journal article and will be made publicly available upon publication of that manuscript. While our analyses include single-subject behavior prediction, they are intended solely for advancing scientific understanding of brain–behavior dynamics and not for clinical or surveillance applications.

## 8 REPRODUCIBILITY STATEMENT

We have taken several measures to ensure the reproducibility of our work. All model definitions, training procedures, and evaluation pipelines are described in detail in the Methods section, with additional hyperparameter choices, optimization details, and ablation studies provided in the Appendix. Preprocessing steps for the fMRI dataset, including voxel selection and behavioral alignment, are fully documented in the supplementary materials. To further facilitate reproducibility, we provide anonymized source code and scripts for running the experiments as supplementary files. Together, these resources should enable readers to reproduce our results and extend our analyses with minimal additional effort.

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

## A    TECHNICAL APPENDICES AND SUPPLEMENTARY MATERIAL

### A.1    EXPERIMENTAL SETUP: SCANNING PROCEDURE

Blood oxygenation level-dependent (BOLD) activity were recorded in a single session consisting of 5 11-minute functional runs. Data were acquired on a 3T Siemens Trio with a 32-channel head coil at the Brain Imaging Center at UC Berkeley. A T2*-weighted gradient-echo EPI sequence with a water-excitation radiofrequency pulse was used to prevent contamination from fat signal (TR = 2.0045 s, echo time = 35 ms, flip angle = 74°, voxel size = $2.24 \times 2.24 \times 3.5$ mm$^3$, field of view = $224 \times 224$ mm$^2$, matrix size = $100 \times 100$, and 30 axial slices to cover the entire cortex). Custom personalized headcases (caseforge, Power et al. (2019)) were used to stabilize the head and to reduce motion artifacts. Anatomical data were also collected to reconstruct the cortical surface (three-dimensional T1-weighted MP-RAGE sequence, $1 \times 1 \times 1$ mm$^3$ voxel size and $256 \times 212 \times 256$ mm$^3$ field of view). Before each functional run, a gradient-recalled echo fieldmap was collected for distortion correction. Respiration and heart rate were recorded using a BIOPAC MP150 system (BIOPAC Systems, Inc.). Gaze location were recorded using an Avotec dark-pupil IR eyetracker at 60 Hz. To ensure accurate eyetracking calibration, at the beginning of every 11-minute functional run, 35 calibration points were presented for 2 seconds each.

### A.1.1    FMRI DATA PREPROCESSING

Freesurfer Dale et al. (1999) was used to reconstruct the cortical surface mesh from the T1-weighted anatomical volumes. The freesurfer anatomical segmentation was checked and manually corrected. Blender (Blender Foundation) and pycortex Gao et al. (2015) were then used to remove the medial wall, and relaxation cuts were then made into each surface. Freesurfer was then used to flatten the cut surfaces. Each functional run was first motion-corrected using the FMRIB Linear Image Registration Tool (FLIRT) from FSL 5.0 Jenkinson and Smith (2001). Next, functional images were unwarped by applying FUGUE from FSL to fieldmaps collected between functional runs. All volumes in the run were then averaged across time to obtain a high-quality template volume. To align data collected across multiple runs, the template volume from the first session was selected as a target, and the template volume from all other runs were aligned to this target. Pycortex was then used to align the functional runs to the anatomical surface. Alignment was checked manually and adjusted as necessary to improve accuracy. Low-frequency voxel response drift was identified using COMPCOR Behzadi et al. (2007) and removed from the signal. Physiological signals from respiration and heartbeats were also regressed out Glover et al. (2000). Voxel activity in each 11-minute run was z-scored separately; that is, within each run, the mean response for each voxel was subtracted and the remaining response was scaled to have unit variance. To remove confounds from the eyetracking calibration sequence and detrending artifacts, the first 35 and last 5 TRs were then discarded from each functional run. For the purpose of this study, the 5 runs were concatenated resulting in 85265 voxel-wise time series with 1475 samples each.

### A.1.2    REGION OF INTEREST (ROI) ABBREVIATIONS

Functional regions were identified using localizer data that was collected separately. Abbreviations for ROIs used in figures are as follows:

- V1: early visual cortex
- IPS: intraparital sulcus
- M1F: primary foot motor area
- S1F: primary foot somatosensory area
- M1H: primary hand motor area
- S1H: primary hand somatosensory area
- hMT+: human middle temporal complex
- FEF: frontal eye field
- SMFA: supplementary foot motor area
- SMHA: supplementary hand motor area

- SEF: supplementary eye field
- PFC: prefrontal cortex.
- SM: supplementary motor cortex
- PM: premotor cortex

### A.1.3 NATURALISTIC BEHAVIOR IN FMRI

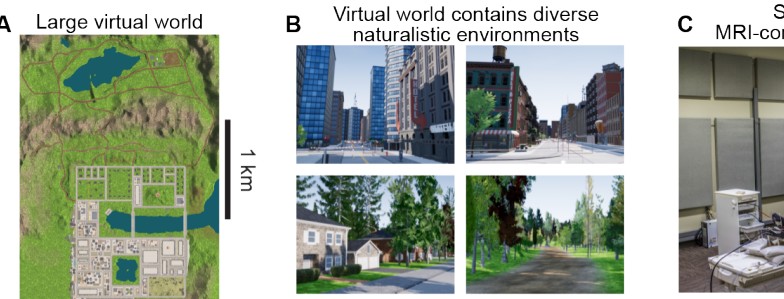

Figure 4: Experimental setup: naturalistic behavior in fMRI. **A** We built a large naturalistic virtual world that spans approximate $2 \times 3$ km. **B** This world contains diverse distinct environments, ranging from urban city to rural forests, and is complete with AI vehicular and pedestrian traffic. **C** The subject used a custom MR-compatible Steer wheel and pedals set to drive through this world.

## B  BENCHMARKING

### B.1  DATA DIMENSIONALITY

To assess the dimensionality of the brain data, we conducted principal component analysis (PCA) and computed the cumulative explained variance as a function of the number of principal components. The results, shown in Fig.5, illustrate how many components are required to capture key variance thresholds in the data. Specifically, 826 components explained 90% of the variance, 975 components explained 95%, and 1,112 components accounted for 99% of the total variance. This analysis informed our choice of dimensionality when applying PCA-based input reductions in the downstream predictive models.

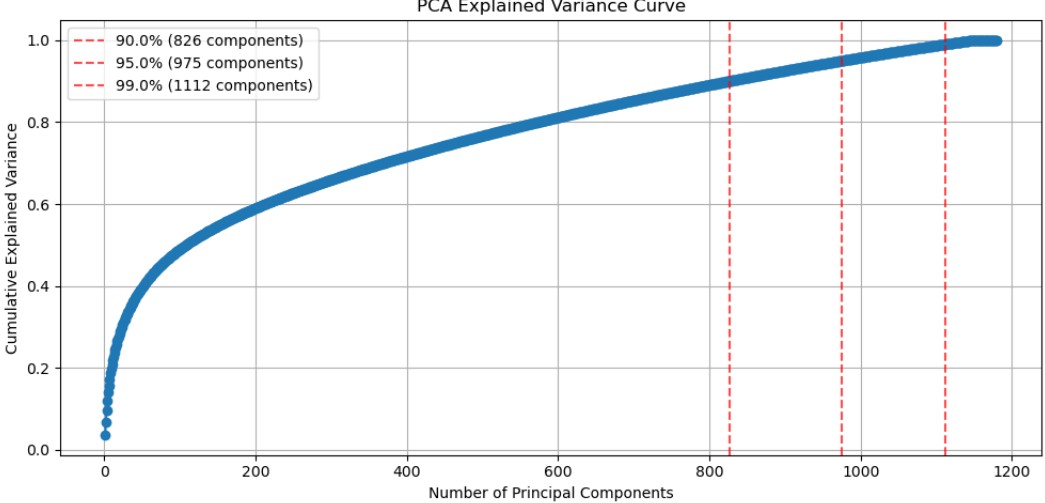

Figure 5: PCA of the voxelwise brain data.

## B.2 Hyperparameters selection

The embedding dimension $E$ and time-delay $\tau$ in MDE are chosen in a fully data-driven manner using the `pyEDM` library. Specifically, we employ pyEDM's EmbedDimension routine, which first scans candidate embedding dimensions $E = 1, 2, \ldots$ and, for each $E$, constructs the corresponding delay-coordinate embedding of the training set. At each $E$, a leave-one-out Simplex projection forecast is performed and the resulting predictive skill (e.g. Pearson correlation or RMSE) is recorded. The optimal embedding dimension is then identified as the smallest $E$ at which forecast skill ceases to improve substantially, ensuring that the attractor is sufficiently unfolded without introducing excess noise or overfitting. This entire procedure is repeated over a range of candidate $\tau$ values, and the $(E, \tau)$ pair that maximizes out-of-sample forecasting performance is selected for all subsequent analyses.

## B.3 Effect of Feature Number on Performance

We evaluated the stability of MDE performance as a function of the number of selected features, testing models with increasing feature counts from 10 to 50. As shown in Figure 6, MDE showed relatively small variation in MAE and RMSE as the number of features increased. Notably, performance improvements plateaued beyond 30 features, indicating that the model achieves strong predictive performance even with a very low number of features. When compared to DL baseline models, at equal dimensionality, MDE demonstrates superior performance compared to all other models tested as we show in Fig.7. These findings suggest that MDE effectively identifies a compact, task-relevant subset of features that prioritizes predictive variance over global variance, making it well-suited for high-dimensional brain data and intrinsically low-dimensional behavioral targets.

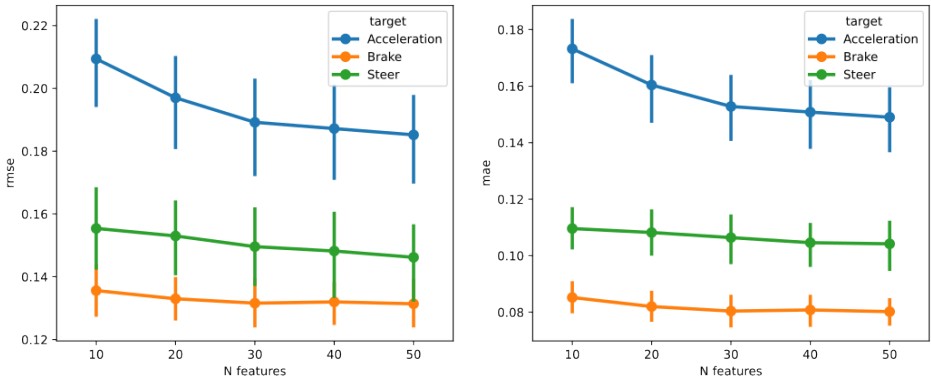

Figure 6: MDE performance varying the number of features. Mean and std values of RMSE (left) and MAE (right) for each target prediction are shown as a function of the number of features used in the MDE algorithm. Targets are color coded.

## B.4 Deep Learning Model Architectures

We compared seven deep learning architectures to MDE as baselines. All models expected inputs of shape $[B, T, F]$ (batch, time, features) and produced outputs of shape $[B, T, 3]$ corresponding to the three behavioral targets. All models were assessed on the reduced dimensions of the inputs, once for PCA with 50 components (small-PCA), and once for PCA with the number of components required for 99% explained variance (large-PCA). Based on the number of components used, parameters for each model were set accordingly. Prior to training, the fMRI data was z-scored and the motor outputs were min-max normalized.

**Non-temporal models.** The multilayer perceptron (MLP) and Bottleneck regressors served as baselines without explicit temporal modeling. The MLP regressor used two hidden layers with 128 and 64 units for the small-PCA case and 256 and 128 units for the large-PCA case. Dropout was set

to 0.2 for small-PCA and 0.3 for large-PCA. ReLU was used for nonlinearities. The Bottleneck regressor had a latent dimension of 16 for small-PCA and 64 for the large-PCA case.

**Recurrent models.** We evaluated gated recurrent units (GRU), bidirectional GRUs (BiGRU), and LSTMs, each with hidden dimension 64 or 128 in the small-PCA and large-PCA regimes, respectively.

**Convolutional model.** The temporal convolutional network (TCN) used 64 channels for the small-PCA and 128 for the large-PCA with kernel size 3 and ReLU nonlinearities between layers.

**Transformer.** The Transformer model used 128 hidden units, 4 attention heads, and 2 layers for the small-PCA and 256 hidden units, 8 attention heads, and 4 layers for the large-PCA. Attention weights were automatically captured via hooks during evaluation.

**Training and evaluation.** We followed a Leave-One-Run-Out (LORO) protocol. At each iteration, one run was held out as the test set, and the remaining four runs were concatenated into a training set. Ten-fold cross-validation was applied to training windows for model selection, after which models were retrained on all training data and evaluated on the held-out run. Each model was trained for 50 epochs and batch size of 32 using Adam (learning rate $10^{-3}$). The procedure was repeated across all five runs, and results are reported as mean and standard deviation across these held-out evaluations. Performance was assessed at the window level using RMSE and MAE.

## B.5 DEEP LEARNING MODELS COMPARISON

The deep learning models used here for benchmarking used PCA-reduced inputs.

When using a number of PC equal to the number of dimensions used in our model (N=50), MDE clearly outperforms all the tested models, as shown in Fig.7. Nevertheless, since we know from the results of PCA that a much higher number of components is needed to explain most of the variance in our data, we also trained and validated the deep learning models using a number of components N= 1180 that explained over the 99% of the variance of the data. In this comparison shown in Fig.8, MDE still shows comparable performance with the other models.

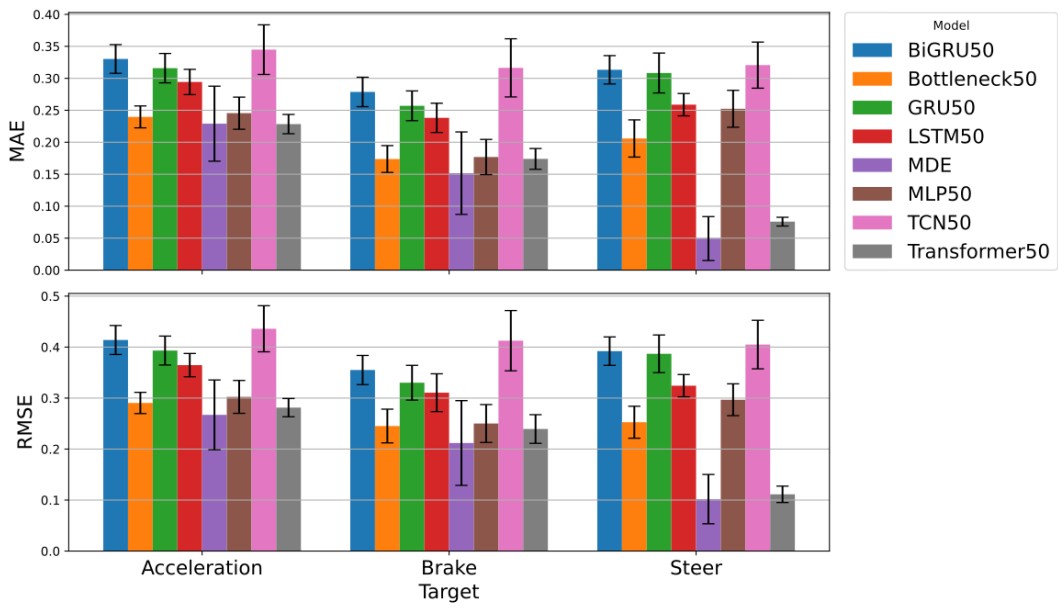

Figure 7: MDE compared to DL models using N=50 principal components.

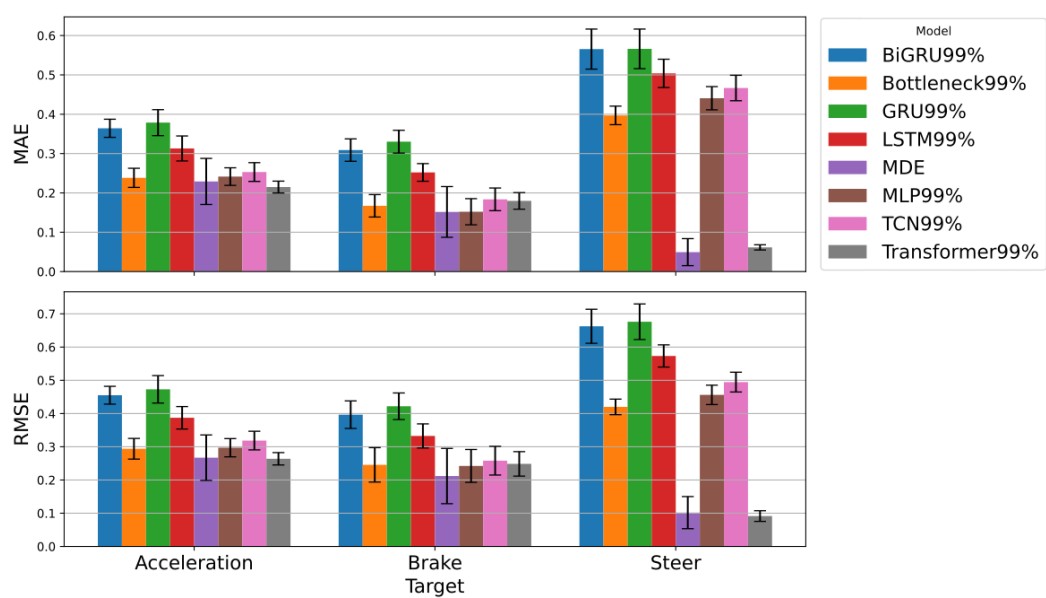

Figure 8: MDE compared to DL using N=1180 principal components explaining over the 99% of the variance of the data.

## C   Network of brain regions identified by MDE for predicting driving behavior

MDE selected a unique set of voxels for predicting driving behavior. The voxel selection by MDE may provide novel insights on the network of brain regions that produce behavior, beyond regions whose activity may be linearly correlated with motor actions. Thus, we examined the distribution of voxels selected by MDE in the brain.

MDE is a parsimonious algorithm that selects a sparse set of voxels. The sparsity of the selected voxels makes the network difficult to characterize for two related reasons. First, the inherent sparsity makes it difficult to identify the extent of functional regions in the brain. Second, because the BOLD signal is noisy, the voxel selection may not be consistent across runs. To overcome both challenges, we identified the set of voxels whose activity is most correlated with the voxels selected by MDE. This set of correlated voxels reveals the extent of the cortical regions selected by MDE as predictive of behavior, and accounts possibly noisy voxel selections.

For **Brake**, MDE selected voxels from the primary foot somatomotor region, secondary motor cortex, the frontal eye fields, supplementary foot motor areas, and the intraparietal sulcus. Additionally, some voxels from the visual periphery are included.

For **Steer**, MDE selected voxels from the primary hand somatomotor regions, secondary motor cortex, supplementary hand motor areas, the right fusiform face area, right posterior lateral parietal cortex, and punctate regions in the dorsalateral prefrontal cortices.

For **Acceleration**, MDE selected voxels from the primari foot somatomotor areas, supplementary motor areas, parietal cortex, hMT, and multiple punctate locations distributed across the prefrontal cortex.

### C.1   Brain Maps of Features Selected by Models

For each behavioral target, we mapped the features selected or the weights of the model onto the cortical surface Fig.9.

**Interpretation of feature maps.**

- **MDE (sparse, causal maps):** Fig. 9 top row shows the voxels identified by MDE as casually predictive of driving behavior. Because the voxels are sparse, in addition to the voxels selected by MDE, we also show voxels whose activity are highly correlated with the selected voxels (thresholded at the 98th percentile). These clusters fall in motor and somatosensory cortices, regions that are known for producing motor outputs, and also some prefrontal regions that may reflect higher-order abstract action planning. By constraining feature selection through dynamical systems principles, MDE yields structured, stable, and neurobiologically plausible maps, consistent with the idea that behaviorally relevant activity resides in low-dimensional manifolds.

- **Lasso (sparse regression):** Fig. 9 second row show voxels identified by lasso regression to predict driving behavior. Like MDE, lasso enforces sparsity, and we also show voxels whose activity are highly correlated with the selected voxels. Lasso enforces sparsity; however, for Acceleration and Steer, it identifies scattered voxels across the cortex that do not appear to be anatomically coherent, and for Brake, it identified voxels only in the hand motor area, and IPS, and precuneus, which do not align with the known function regions that produce motor behavior. Because it ignores temporal dynamics and causality, Lasso likely selected voxels that are by chance correlate with behavior, rather than mechanistic substrates of specific actions. This instability undermines interpretability despite apparent sparsity.

- **Ridge regression (distributed weights):** Fig. 9 third row shows weights from ridge regression. Ridge produces diffuse weight maps across the whole brain. Notably, these weights are noisy and do not appear to identify any consistent regions. While such distributed and spatially noisy weights can achieve predictive accuracy by exploiting correlated activity, they lack anatomical specificity, making them neurobiologically implausible as explanations of behavior.

- **Transformers (attention-based models):** Fig. 9 rows four and five show weights from transformer models. To interpret Transformer models, we derive voxelwise contribution maps. For each left-out run, a single PCA is fit on $z$-scored train+validation data to define a locked basis. Models are trained on either the first 50 components (PCA–50) or the smallest number $K$ explaining $\geq 99\%$ variance (PCA–99%). After training, we compute per-output attributions (Acceleration, braking, Steer) using Integrated Gradients with a zero baseline in PC space, aggregate over time and test batches to obtain a component-importance vector, and back-project to voxels using the same PCA loadings ($w_{\text{vox}} = C^\top w_{\text{PC}}$). Maps are $L_1$-normalized, sign-aligned across folds, and averaged. These maps show widespread weights gradients across cortex. While these weights are anatomically coherent (unlike ridge weights), they are distributed across the cortex and do not parsimoniously identify predictive regions. Furthermore, they are unstable across PCA dimensionalities: maps from PCA–50 and PCA–99% often differ substantially, with regions even flipping sign. This instability indicates that Transformer attributions reflect sensitivity to preprocessing choices rather than robust mechanistic substrates.

In summary, unlike Lasso, Ridge, or Transformers, MDE integrates causal feature selection with state-space reconstruction, yielding stable, interpretable maps localized to motor and sensory cortices. This provides mechanistic insights consistent with established neurobiology while achieving predictive performance comparable to or exceeding black-box models.

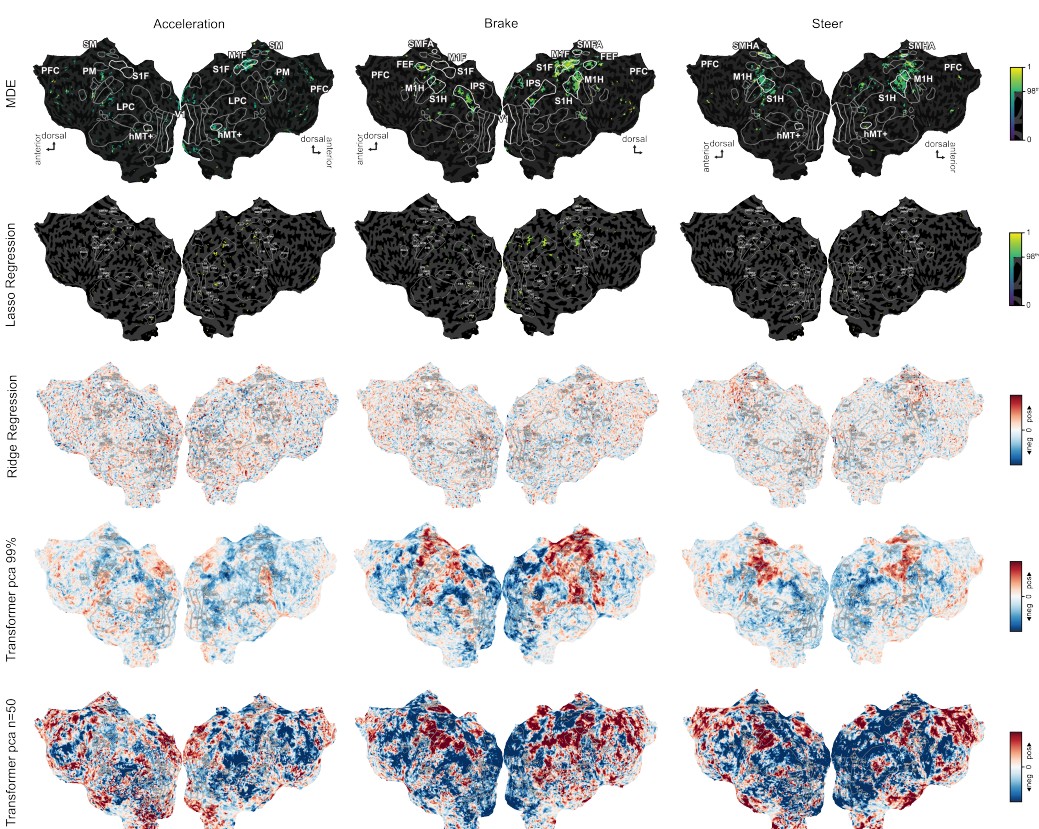

Figure 9: Voxel maps for decoding Acceleration, Braking, and Steer behaviors from fMRI. Each column corresponds to one behavioral dimension (Acceleration, Brake, and Steer), and each row to a predictive model. For sparse models (MDE, Lasso), plots show voxels directly selected by the model and also with voxels whose activity are highly correlated with them (thresholded at the 98th percentile). For distributed models (Ridge, Transformers), maps show signed voxel weights or importance values (red = positive, blue = negative). Results show that lasso regression, while sparse, does not identify any anatomically coherent regions; ridge regression, while able to predict behavior, do not identify any plausible functional regions that drive activity; transformers, while also able to predict activity, are sensitive to preprocessing parameters, suggesting that they do not actually capture the true brain substrates of behavior

We investigated whether restricting the input to only the voxels that are causally linked to the target behaviors would improve regression performance. To identify these voxels, we computed causal relationships using Convergent Cross Mapping (CCM) between each voxel's time series and the three behavioral variables. We then masked the data to retain only the causal voxels and trained Ridge and Lasso regression models on this reduced representation. We did not analogously restrict Transformer inputs to CCM-identified voxels, because the model is trained in a locked PCA basis that aggregates distributed cortical information, and masking would alter this basis and remove global context. Also, the Transformer's nonlinear, attention-based architecture can in principle learn to emphasize behavior-relevant signals without an explicit causal mask. The results in Table **??** showed that the performance of both models on the causal-masked data was nearly identical to their performance when trained on the full set of brain voxels. Maps of selected features are shown in Fig. 10.

When restricted to only casual voxels, ridge regression weights shows positive mappings between distributed regions in supplementary motor cortex, premotor cortex, parietal cortex, and the visual periphery to Acceleration; the primary foot motor cortex to Brake; and primary hand and foot motor, supplementary motor areas, and frontal eye fields, and precuneus and lateral parietal cortex regions

Table 3: MAE

| model
target | lasso | lasso_masked | ridge | ridge_masked |
|---|---|---|---|---|
| Acceleration | 0.151 ± 0.017 | 0.149 ± 0.016 | 0.144 ± 0.019 | 0.143 ± 0.019 |
| Brake | 0.08 ± 0.003 | 0.081 ± 0.003 | 0.081 ± 0.002 | 0.081 ± 0.003 |
| Steer | 0.116 ± 0.007 | 0.116 ± 0.006 | 0.122 ± 0.004 | 0.122 ± 0.004 |

Table 4: RMSE

| model
target | lasso | lasso_masked | ridge | ridge_masked |
|---|---|---|---|---|
| Acceleration | 0.184 ± 0.02 | 0.183 ± 0.019 | 0.178 ± 0.023 | 0.177 ± 0.023 |
| Brake | 0.129 ± 0.009 | 0.13 ± 0.01 | 0.126 ± 0.009 | 0.126 ± 0.009 |
| Steer | 0.159 ± 0.012 | 0.159 ± 0.013 | 0.168 ± 0.012 | 0.167 ± 0.012 |

to Steer. These regions are largely consistent with the regions identified by MDE (3), albeit more noisy and distributed. On the same casual voxels, lasso regression selected voxels in the anterior RSC, anterior OPA, hMT+, and lateral parietal cortex for Acceleration, Brake, and Steer. While these voxel selections are more spatially coherent than results from training on all voxels, they do not correspond well to those selected by MDE. These results suggest that restricting the pool of voxels to casual voxels may improve the neurobiological plausibility of ridge regression, but not lasso regression, to predict behavior.

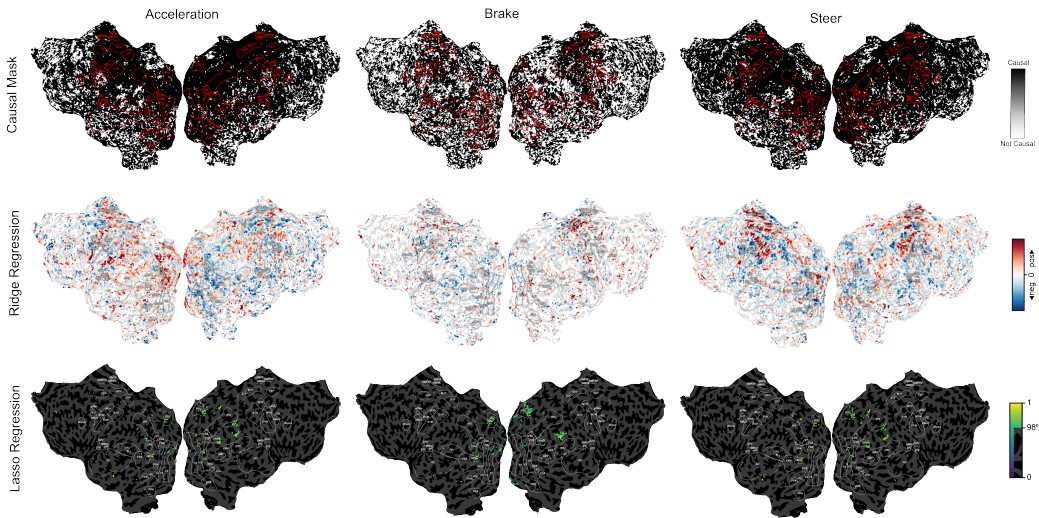

Figure 10: Voxel maps for decoding Acceleration, Braking, and Steer behaviors from causal voxels only. Each column corresponds to one behavior, the first row shows the causal mask (black voxels are causal to the behavior, white voxels are not) and the second and third rows correpsond to the results obtained with Ridge and Lasso regression, respectively. For ridge regression, red–blue maps show signed voxel weights or importance values (red = positive, blue = negative). For Lasso regression, green–yellow overlays indicate voxels directly selected by the model together with voxels highly correlated with them (thresholded at the 98th percentile).

## D  GENERALIZED TAKENS THEOREM

Takens' original embedding theorem (Takens, 1981b) states that for a smooth dynamical system $(M, \phi^t)$ on a compact $d$-dimensional manifold $M$, the delay-coordinate map constructed from a generic smooth observable $h : M \to \mathbb{R}$,

$$F(x) = \big(h(x),\, h(\phi^\tau(x)),\, \ldots,\, h(\phi^{(m-1)\tau}(x))\big),$$

is an embedding of $M$ into $\mathbb{R}^m$ provided $m \geq 2d + 1$. Deyle and Sugihara (Deyle and Sugihara, 2011) generalized this result by allowing reconstruction from a collection of measurement functions or a combination thereof rather than delays of a single observable. Specifically, given a set of smooth measurement functions $h_i : M \to \mathbb{R}$, the mapping

$$F(x) = \big(h_1(x),\, \ldots,\, h_p(x)\big),$$

is an embedding for generic choices of $\{h_i\}$ if the joint embedding dimension $m = \sum_{i=1}^p m_i$ exceeds $2d$. This "generalized embedding" guarantees that the reconstructed attractor is diffeomorphic to the original system's attractor, enabling faithful recovery of system dynamics from multivariate or heterogeneous time series or any combination thereof.

## E  CAUSALITY IN DYNAMICAL SYSTEMS

Convergent Cross Mapping (CCM), introduced by Sugihara et al., is a method for detecting causal relationships in dynamical systems using state-space reconstruction as formalized by Takens' theorem (Takens, 1981b). The key idea is that if variable $X$ drives variable $Y$, then the historical states of $Y$ should contain a footprint of $X$'s dynamics, allowing $X$ to be predicted from $Y$'s reconstructed attractor.

To test whether $X \to Y$, CCM reconstructs the attractor from the $Y$ time series and uses simplex projection to predict $X$. The central diagnostic is *convergence*: as the observation density increases, prediction accuracy improves and eventually stabilizes. This occurs because longer recordings or denser maps produce denser attractors, which yield better nearest-neighbor estimates. Convergence thus distinguishes genuine causal influence from spurious correlation: if $X$ truly drives $Y$, then information about $X$ will be recoverable from $Y$'s embedding in a consistent, map density-dependent manner. In our framework, we use this convergence property as a principled criterion for feature selection: variables are included only if their embeddings demonstrate convergent predictability of the target, ensuring that selected features exert a directional influence on behavior.

## F  BROADER IMPACT

Our work on MDE not only advances dynamical-systems decoding in neuroscience but also generalizes to any multivariate time series system where both accurate forecasting and mechanistic insight are prized. Neural decoding has applications in clinical settings, where shifts in the geometry of patient-specific neural manifolds or changes in short-horizon forecast error profiles may serve as sensitive, interpretable biomarkers for brain disorders and injuries, enabling earlier diagnosis and more precise monitoring of treatment response. For brain–computer interfaces, our sparse, causally validated feature selection combined with accurate, short-term behavioral forecasting can enhance prosthetic control, while providing clear mechanistic insight into the neural circuits at play. Finally, by translating neural or behavioral signals into low-dimensional state spaces, MDE can inform adaptive robotics policies paving the way for personalized, closed-loop technologies across health, safety, and human–machine interaction. MDE can also offer insights in realms different than neuroscience. For example, in genomics, our method can reveal causal gene–gene interactions and predict expression dynamics under perturbations. In ecology, it can reconstruct population attractors from species count or environmental data, forecasting regime shifts while identifying key drivers. In epidemiology, it could model outbreak trajectories from infection and mobility time series, pinpointing variables that causally influence transmission. Beyond biology, applications also include financial markets (forecasting asset trajectories and uncovering leading indicators), and smart infrastructure (predicting energy demand and identifying critical load factors). By recovering low-dimensional attractors and selecting causally relevant features, MDE offers a unified, interpretable framework for prediction and discovery across science, engineering, and policy.

## G   Computational Resources

The experiments were carried out on a high-performance Linux server equipped with two AMD EPYC 7742 64-core CPUs (128 cores total at 1.5 GHz), 1 TB of RAM, and an NVIDIA A100 PCIe 40 GB GPU. Under this configuration, running MDE on each cross-validation fold required roughly 1.2 hours (about 1 hour and 12 minutes) per fold to complete. The linear regression models take on average 0.1 ms for a single point prediction.

## H   LLM usage

Portions of the manuscript text were refined with the assistance of a large language model (LLM), which was used solely for editing and polishing writing style. All technical content, analyses, and conclusions were generated and verified by the authors.

