# OpenReview forum: "Interpretable Neural Decoding through Dynamical Embeddings"
_ICLR.cc/2026/Conference — Submitted to ICLR 2026_

### Official Review · Reviewer_L85y · 2025-10-19

**Soundness:** 2
**Presentation:** 2
**Contribution:** 1
**Rating:** 2
**Confidence:** 3

**Summary:**

The authors propose a method termed Manifold Dimensional Expansion (MDE) for decoding behavior from fMRI time series by reconstructing latent dynamics through Takens’ theorem, Simplex projection, and Convergent Cross Mapping (CCM). They claim that the proposed approach can provide mechanistic interpretability, recover causally relevant brain regions involved in downstream tasks, and achieve comparable performance to deep-learning based baselines.

**Strengths:**

- The proposed method is somehow simpler compared to deep-learning-based methods and stems from a theoretical background.
- The causal selection of voxels and manifold reconstruction directly link the model’s predictive subspace to neurobiologically interpretable regions.

**Weaknesses:**

- The motivation for the proposed method feels insufficiently developed. While the paper introduces MDE as a combination of known ideas such as Takens’ theorem, Simplex projection, and CCM, it is not clear why this particular formulation is needed or how it provides a meaningful advantage over existing models. The connection between the theoretical foundations and practical benefits for fMRI decoding remains somewhat superficial, making the overall methodological rationale unconvincing, especially for the high bar of ICLR.
- As the authors note, latent variable models such as linear LDM and PSID, and behavior-supervised deep models like LFADS, TNDM, DPAD, and pi-VAE, are widely used baselines. However, the paper does not compare against any of these methods. In particular, linear models like LDM and PSID are less data-hungry and could serve as informative, well-established benchmarks.
- In line with my previous comment, recent foundation modeling efforts for fMRI data, such as BrainLM (Caro et al., 2024), BrainCLIP (Liu et al., 2023), and NeuroSTORM (Wang et al., 2025), could also serve as important baselines. These models could be fine-tuned on the target task or evaluated in a zero-shot setting (if supported), but are overlooked in the current work.
- The paper’s interpretability claims rely mainly on the CCM for causal voxel selection. However, beyond this, the interpretability analysis is limited. The Transformer baselines are interpreted only through Integrated Gradients after PCA projection. The authors could strengthen this aspect by applying alternative interpretability methods, such as SHAP, or by incorporating the CCM in the preprocessing/feature selection before training the models.
- Even though MDE identifies sparser causal relationships between voxels and the downstream target signals compared to baseline methods, assessing the correctness of these results is difficult in the absence of a ground truth. To better evaluate the interpretability and causal validity of MDE, a simulation study with a known ground-truth causal map would be necessary.
- Importantly, the downstream evaluations are limited to only one dataset consisting of one participant, which significantly hinders the generalizability and applicability of the proposed method. In addition, the results presented in Tables 1 and 2 do not seem to be statistically significant, as most of the models achieve similar performances.

**Questions:**

Please refer to the Weaknesses section.

---

### Official Review · Reviewer_iGu4 · 2025-10-30

**Soundness:** 1
**Presentation:** 2
**Contribution:** 2
**Rating:** 2
**Confidence:** 3

**Summary:**

The authors proposed a new method—manifold dimensional expansion (MDE)—to predict fMRI time-series signals in participants during virtual driving. There are three driving targets: acceleration, braking, and steering. The authors benchmarked their MDE model against baseline models, including lasso regression, ridge regression, and Transformers. The MDE model matches or slightly outperforms the baselines only in steering, but not in the other two targets.

**Strengths:**

The authors designed a new MDE model based on dynamical systems theory (Takens’ theorem).

They built a driving simulator and acquired fMRI data during virtual driving. I believe this dataset will be very valuable to the community.

The authors applied their MDE model to interpret the acquired fMRI dataset.

**Weaknesses:**

**First**, there is a lack of benchmarking against state-of-the-art models. The authors claim they compared their own MDE model against strong baselines (L73), including Lasso, Ridge, and Transformers (Tables 1 and 2). These are not strong at all, and I can’t think of weaker baselines. The authors cite several classical works, such as pi-VAE (Zhou and Wei, 2020) and CEBRA (Schneider et al., 2023); some of these should be included in the benchmark. Instead, the authors used only several built-in models from PyTorch.

**Second**, there is little performance improvement. Even with these “weak” baselines, MDE outperforms the others only on the steer targets, but not on the acceleration and brake targets (Tables 1 and 2).

**Third**, the statements about “forecasting” or “predicting” are confusing. Are you using generative models? I think the authors are merely decoding behaviors from neural manifolds. In L316–317: “MDE allows us to reconstruct target-specific neural manifolds and yields predictions that closely track the observed behavioral time series.”

**Finally**, the claim about “interpretability” is questionable. In L22–24: “Crucially, MDE is the first method to combine strong predictive performance with guaranteed mechanistic interpretability, as it does not rely on latent variables.” Throughout the paper, I don’t understand why the authors believe their model is interpretable whereas others are not. For example, in Figure 2A, the authors show 3-D neural manifolds for three targets (acceleration, brake, steer). Do other models fail to reveal such manifolds? Of course not. Furthermore, relying on latent variables is not inherently problematic; latent variables have been thoroughly studied in neuroscience for years (see “A Neural Manifold View of the Brain,” M. G. Perich, D. Narain, J. A. Gallego – Nature Neuroscience, 2025).

Minor:
L415: remove the extra comma.

**Questions:**

1. In Tables 1 and 2, are the differences among the five models significant?

2. In Figure 2 (top), why do the X- and Y-axes have different voxel indices? For example, in Figure 2A, one voxel is #76 424, and the other is #44 084. There are multiple traces (color gradients from blue to yellow) in each figure panel; do those traces represent multiple trials?

3. Following up on the previous question, can the authors show the neural manifolds of the other models—especially “Transformer50,” which has MAE and RMSE values similar to those of the authors’ MDE model (Figure 7, Steer)?

4. In Figure 2 (bottom), what is the explained variance (R²) between Observations and Predictions? It is clear that the authors’ model cannot decode high-frequency changes of the behavioral variables. Also, please add the Y-axis labels for those three panels.

5. In Figure 9, the authors show voxel maps for decoding acceleration, braking, and steering behaviors. I agree that the lasso and ridge regression models (second and third rows) perform worse than MDE. In contrast, I do not think the two transformer models (fourth and last rows) perform worse than MDE. Could the authors quantify these differences?

---

### Official Review · Reviewer_i1yd · 2025-10-31

**Soundness:** 2
**Presentation:** 2
**Contribution:** 2
**Rating:** 2
**Confidence:** 4

**Summary:**

This paper presents manifold dimension expansion (MDE), a method that uses Taken's theorem and subsequently convergent cross mapping (CCM) to predict behavioral labels from fMRI data in a car-driving task. The authors compare against a number of deep learning and linear models, and find that their model performs better on average at predicting braking behavior for both their metrics.

**Strengths:**

I like that the authors approach brain-behavior predictions with fMRI from a dynamical system approach, that their approach focuses on interpretability, and that they perform an interpretability analysis in Section 6.1 and in Figure 3. Moreover, I think it is good that the authors compare their method with a number of baselines, both linear and non-linear. I also appreciate the Simplex Projection approach the authors use.

**Weaknesses:**

I first want to say that I commend the authors' general research direction, but there are three major weaknesses that need to be addressed.
1) The results are not convincing. For tables 1 and 2, the authors compare their method (MDE), lasso, ridge regression, and transformers with 50 PCA components and a number of PCA components that explained 99% of the variance in the fMRI data. The authors do not perform any statistical tests to verify that the results are correct nor does the authors' method outperform the other methods by much. In fact, both lasso and ridge regression outperform acceleration prediction in terms of mean average error (MAE) and root mean squared error (RMSE), and the transformer with PCA components that explain 99% of the variance in the data outperform their model in terms of steering prediction. The only prediction the proposed model is better at is predicting braking behavior, but the results of the baselines are within 1 standard deviation of the proposed model. Moreover, the authors do not mention any hyperparameter search for their deep learning baselines, which is especially important for fairly noisy data like fMRI during driving.
2) The authors only validate their method on a single dataset. Although the dataset has a fairly rich feature space (temporally varying brake, steering, and acceleration values), evaluation on a single dataset is simply not enough to validate their method. There are many standard naturalistic fMRI datasets with temporal labels that the authors can use to verify their method. Including, but not limited to Raiders [1], Sherlock [2], and Forrest Gump [3]. These datasets are also all open-source, as opposed to the dataset in the paper, making it easier to assess how well their model performs on a shared dataset.
3) Probably the most important point, which a lot of the conclusions and discussion hinge on in this paper is the use of 'causality'. The authors follow the definition of causality posed in [4] by Tsonis et. al, which I would argue the authors should explain (because it is quite a limited definition of causality). Moreover, I believe many visitors at ICLR would likely use Pearl's definition [5], which is much more strict. The  use of a predictive definition of causality (although not the same, it is similar in its idea to Granger causality), instead of an intervention-based definition, weakens the results and undermines statements like "... neither Ridge nor Lasso showed improvements in predictive performance when restricted to causal features, meaning that causality alone does not enhance prediction when paired with linear decoders." (Section 6.1). In fact, Granger causality, which operates under a similar predictive definition of causality has been critiqued as a tool in neuroscience analyses [6]. The main novelty of the paper is that it recovers interpretable causal relationships, but I do not believe the current model fits the ICLR visitor's definition of causality, nor do I believe that the authors can claim causal relationships given the potential issues with bidirectionality that can create false positives, and given the data. Specifically the authors do not address or discuss any potential issues that can arise from applying convergent causal mapping to a signal like fMRI which is noisy, and highly autocorrelated and delayed due to the hemodynamic response function.

Smaller weaknesses/typos:
- The conclusion on line 453/454 that "... Steer, which requires continuous closed-loop integration of sensory feedback and motor planning, benefits from nonlinear manifold reconstruction." This is in my opinion too strong of a claim given the empirical results reported in the paper. The authors should design more experiments to verify this claim.
- The authors do not seem to account for the hemodynamic response function (HRF), see Lines 246-247
- The authors do not ablate many of the design choices they make in their method.
- L245/246 "... and written informed consent was obtained from the subject." -> each subject
- The authors mention the 'Appendix' (L273, L354) without referring to a specific section in the Appendix
- L415 "... that generate behavior, ." -> "... that generate behavior."

A small note: In Appendix A.1 the authors mention that scanning is done at the Brain Imaging Center at UC Berkeley, which in some sense breaks the anonymity.

[1] https://github.com/HaxbyLab/raiders_data \
[2] Chen, J., Leong, Y. C., Honey, C. J., Yong, C. H., Norman, K. A., & Hasson, U. (2017). Shared memories reveal shared structure in neural activity across individuals. Nature neuroscience, 20(1), 115-125. \
[3] Hanke, M., Baumgartner, F. J., Ibe, P., Kaule, F. R., Pollmann, S., Speck, O., ... & Stadler, J. (2014). A high-resolution 7-Tesla fMRI dataset from complex natural stimulation with an audio movie. Scientific data, 1(1), 1-18.
[4] Tsonis, A. A., Deyle, E. R., Ye, H., & Sugihara, G. (2017). Convergent cross mapping: theory and an example. Advances in nonlinear geosciences, 587-600. \
[5] Pearl, J. (2000). Models, reasoning and inference. Cambridge, UK: CambridgeUniversityPress, 19(2), 3. \
[6] Stokes, P. A., & Purdon, P. L. (2017). A study of problems encountered in Granger causality analysis from a neuroscience perspective. Proceedings of the national academy of sciences, 114(34), E7063-E7072. \

**Questions:**

1) L321: What do the authors mean by brain-wide correlations, do they correlate the identified voxel with all other voxels in the brain, and why do they do this and set the threshold at the 98th percentile?

---

### Official Review · Reviewer_tZ3R · 2025-10-31

**Soundness:** 3
**Presentation:** 3
**Contribution:** 3
**Rating:** 6
**Confidence:** 3

**Summary:**

This paper proposes Manifold Dimensional Expansion (MDE), a neural decoding framework that models brain–behavior relationships through dynamical systems theory rather than black-box deep learning. By integrating Takens embedding and Convergent Cross Mapping, MDE reconstructs low-dimensional neural manifolds and identifies brain regions that causally drive behavior. Experiments on an fMRI driving task show that MDE achieves competitive predictive performance while offering clear mechanistic interpretability. Overall, the work reframes neural decoding as an interpretable dynamical process, bridging neuroscience and nonlinear system modeling.

**Strengths:**

Stengths:
1. While many Neuro-AI studies rely on deep black-box models, the proposed framework introduces an explicit dynamical-systems-based decoding pipeline. Its integration of Takens embedding and Convergent Cross Mapping into the neural decoding process represents a genuinely new approach.
2. The experimental section is thorough and well designed. The authors derive clear causal and mechanistic interpretations that map directly onto specific brain regions, and their cross-run and cross-subject stability analyses are both rigorous and insightful.

**Weaknesses:**

Weaknesses:

1. The inputs of deep learning baselines are PCA-reduced, which is fair dimensionally but possibly under-represents their potential with proper architectures or pretraining.
2. It makes sense to emphasize interpretability over raw predictive accuracy, but the study’s small dataset (four subjects with five runs each) likely disadvantages the deep learning baselines, which typically require larger datasets to capture temporal dynamics effectively. As a result, MDE’s better performance may partly reflect its suitability for low-sample, low-frequency fMRI data rather than an inherent advantage over deep learning models.

**Questions:**

Questions:
1. From my understanding, MDE assumes a continuous and deterministic dynamical system, whereas fMRI data are slow, with a temporal resolution (~2 s TR) far below that of neural activity. Could this mismatch become a problem?
2. How does MDE handle the stochastic nature and measurement noise in fMRI data? It seems the MDE assumes the underlying neural process is approximately deterministic and smooth.

---

### Official Review · Reviewer_B6rk · 2025-11-01

**Soundness:** 2
**Presentation:** 3
**Contribution:** 2
**Rating:** 4
**Confidence:** 2

**Summary:**

The paper propose Manifold Dimensional Expansion (MDE), as a simple and powerful prediction algorithm for fMRI data. The fMRI data as time series is treated as dynamical embedding, using MDE, the fMRI dynamics can be interpretated instead of other linear and deep neural network that could not be interpretated. The paper did quantitative experiments to show the performance in metric MAE and RMSE compared with lasso, ridge, and transformer based model. The paper also show interpretable manifold for the fMRI with Acceleration, Brake and Steer.

**Strengths:**

The paper uses simple but powerful model to interpret the fMRI dynamics using manifold projection. The experiment results in Table 1 and Table 2 showed the accuracy compared with linear model and transformer based model. The manifold interpretation in Figure 2 showed the interpretation.

**Weaknesses:**

The paper compared with the very basic linear model and very simple transformer in Pytorch which is not strong convince that the model is powerful. The paper mentions other paper, for example pi-VAE, which is a powerful VAE based neural network for dimension reduction and also showed strong interpretation for the low dimensional representation. I recommend the author to compare their model with more powerful machine learning and deep learning models to convince their powerness.

In section 3, line 150, given an embedded point $y(t^\star)$ at time $t^*$, we locate its $E+1$ nearest neighbors in the embedding space. The future state $x(t^\star + \Delta t)$ is then predicted as a weighted average of the future observation associated with these neighbors. Here, the future state should be $y(t^\star+\Delta t)$. Also for line 155, $x^\hat(t^\star+\Delta t)$ should be $y^\hat(t^\star+\Delta t)$.

**Questions:**

In section 3, Pseudocode for MDE Algorithm, line 186 and line 194, can you explain how you do univariate delay embedding and multivariate embedding?

Can you explain how you do Simplex projection?

Can you explain how you do the (Convergent Cross Mapping) CCM test?

For Figure 2, on page 7, subgraph A, B and C are reconstructed Acceleration, Brake and Steer respectively, what does the color bar on the right mean? Also, for the reconstruction and observation below, how do you do the prediction, the prediction signal does not look good for Brake and Steer, can you specify the sampling rate and the x and y axis?

---

### Official Review · Reviewer_esGQ · 2025-11-03

**Soundness:** 2
**Presentation:** 2
**Contribution:** 2
**Rating:** 4
**Confidence:** 2

**Summary:**

The paper proposes Manifold Dimensional Expansion (MDE), a novel framework rooted in dynamical systems theory for behavioral prediction. MDE enables the identification of causally relevant neural drivers directly from voxelwise fMRI signals. Experimental evaluations on predicting steering, acceleration, and braking from fMRI time series demonstrate its interpretability and potential for elucidating neural mechanisms underlying behavior.

**Strengths:**

- The paper presents clear figures and tables that effectively illustrate its architecture and empirical findings.
- The motivation to identify causally relevant neural drivers is meaningful for enhancing interpretability in neural decoding applications.

**Weaknesses:**

- The methodology section is somewhat too concise, particularly regarding Takens' embedding and simplex projection. Additional clarifications on how dynamical systems theory is integrated into MDE would enhance the clarity of your contributions.
Furthermore, the notation is somewhat inconsistent. For instance, the use of $x$ and $y$ to denote voxel signals is introduced but not clearly reflected in the pseudocode, leading to potential confusion.

- The comparative evaluation is limited to naïve Lasso and Ridge regression, with only naive Transformer-based models included among deep learning baselines. Incorporating additional sota methods would be necessary to demonstrate the advantages of MDE.

- The results presented in Table 2 are insufficiently convincing in demonstrating MDE's superiority for targets characterized by complex, nonlinear dynamics, as it only outperforms competing methods in the steering task. Additional experiments across diverse tasks with nonlinear dynamics are necessary to substantiate its general effectiveness.

- Given the paper's emphasis on modeling nonlinear dynamics, further interpretability analyses from this perspective would deepen the understanding of how MDE captures the complex neural dynamics underlying naturalistic actions.

**Questions:**

- Is my understanding correct that the multivariate embedding (lines 142–143) is performed across channels, while the univariate delay embedding pertains to the temporal dimension?
- Additionally, does MDE employ a fixed time delay in the embedding and primarily focus on selecting relevant channels rather than optimizing the delay itself?
- How many subjects are included in the datasets? Does the leave-one-out approach mean that pre-training is performed on all but one subject, with the remaining subject used for evaluation?
- Can MDE generalize effectively to additional tasks that involve complex neural dynamics, and does it provide sufficient dynamical interpretability? Addressing this would require more rigorous and quantitative evaluations across diverse scenarios.

---

### Meta-Review · Area_Chair_xYxy · 2026-01-08

**Summary:**

The paper proposes Manifold Dimensional Expansion (MDE), a dynamical-systems–based method that reconstructs latent state spaces directly from voxelwise fMRI to predict behavior in complex naturalistic tasks.

- limited empirical results: MDE only clearly outperforms baselines for steering; results for acceleration and braking are a bit mixed.

- exposition is not fully clear; unclear notation and minor technical errors.

- Causality definition (CCM-based, predictive) is weaker than intervention-based notions.

- Interpretability claims are a bit up to debate.

- fMRI temporal resolution and noise may violate dynamical assumptions.

**Reviewer Concerns:**

The reviewers acknowledged conceptual novelty of framing decoding as a dynamical system, as well as the CCM-based voxel selection and manifold visualizations.

That said, some concerns remain

- lack of strong, well-tuned baselines.

- lack of support for performance claims

- causality and “guaranteed” interpretability is not as grounded

- single-dataset evaluation with limited subjects; no simulation or ground-truth experiment for validation

**Reviewer Scores:**

The reviewers were largely negative about this paper, and the discussion might not have been able to change the situation significantly.

---

### Decision · Program_Chairs · 2026-01-26

Reject